# The History of the Intestinal Microbiota and the Gut-Brain Axis

**DOI:** 10.3390/pathogens11121540

**Published:** 2022-12-15

**Authors:** Zuzanna Lewandowska-Pietruszka, Magdalena Figlerowicz, Katarzyna Mazur-Melewska

**Affiliations:** Department of Infectious Diseases and Child Neurology, Poznan University of Medical Sciences, 60-830 Poznan, Poland

**Keywords:** microbiota, gut-brain axis, history of medicine

## Abstract

The gut-brain axis and the intestinal microbiota have been an area of an intensive research in the last few years. However, it is not a completely novel area of interest for physicians and scientists. From the earliest centuries, both professionals and patients turned their attention to the gastrointestinal system in order to find the root of physical and mental disturbances. The approach to the gut-brain axis and the therapeutic methods have changed alongside the development of different medical approaches to health and illness. They often reflected the social changes. The authors of this article aim to provide a brief history of the gut-brain axis and the intestinal microbiota in order to demonstrate how important the study of these systems is for both scientists and medical professionals, as well as for the general public. We analysed the publications accessible through PubMed regarding the microbiota and gut-brain axis history. If available, we accessed the original historical sources. We conclude that although the history of this science might be long, there are still many areas that need to be researched, analysed, and understood in future projects. The interest in the subject is not diminishing, but rather it has increased throughout the years.

## 1. Introduction

The human microbiome and its connection with the central nervous system has recently become a highly popular subject. The influence of bacteria on our emotions or behaviour was observed and led to discovery of the so-called gut-brain axis [1]. The neuroactive substances secreted by the bacteria, such as neurotransmitters, short chain fatty acids or amino acids, might influence our behaviour, emotions, cognition, and pain management [2].

Our ancestors observed that there was a link between digestion and emotions, mood, and behaviour [3]. The faecal transplant has been used in Chinese traditional medicine since the Dong-jin dynasty and in Ancient Greece [4,5]. The transplantation was also researched in 17th century by Fabricius Acquapendente [6]. We can observe usage of milk products as a remedy for gastrointestinal problems in the works of Hippocrates, Avicenna, Galen, and Pliny the Elder, and the Bible [7].

Moreover, 18th century vitalists were the ones who began to look at the possibility of an intestinal-brain axis in a more systemic way [8]. In the 19th century, physicians tended to blame the gastrointestinal system for its influence on mind [1,9]. In the 20th century, scientists approached the issue with the opposite perspective, believing that it was the mind that was responsible for abnormalities in gut activity [10,11,12]. In the 21st century, modern methods of research have helped to revive the subject of the gut-brain axis and the role of microbiota in this system [13,14].

This paper aims to summarise the broad history of the gut-brain axis and the intestinal microbiota. We should not overlook the fact that we have made great progress in this field of research and that we still cannot fully predict how much further discovery is still awaiting us in this continually evolving research area.

## 2. Materials and Methods

We used the PubMed search engine, accessible on https://pubmed.ncbi.nlm.nih.gov/, to access the articles on microbiota and gut-brain axis history (accessed on 2 June 2022). We took into consideration the results matching the search engine query “(microbiome) OR (microbiota) OR (gut-brain axis) AND (history)”.

Next, we manually analysed the papers to present a short review on the history of intestinal microbiota and the gut-brain axis.

## 3. Results and Discussion

In the PubMed engine, we obtained 2908 results, dating from 1976 to 2022. In Figure 1 we present the growing number of the articles published on the analysed subject throughout the years.

### 3.1. The Evolution of the Microbiota

The evolution of the microbiota itself is a very intriguing subject. There are significant differences in an individual’s gut throughout their life and between individuals; however, we can observe some patterns in the population [14,15,16,17].

The most published research, including the Human Microbiome Project (HMP) or METAgenomics of the Human Intestinal Tract (MetaHIT), concentrates on research groups from highly industrialised populations [13,14,15]. From smaller research projects, we already know that the microbiota composition of the “non-Western” populations is different from that of the Western populations [18,19,20,21,22,23,24,25]. The diet in industrialised populations is higher in carbohydrates, fat and meat, while the amount of fibre intake is lower than in “non-Western” countries [26]. There are differences in hygiene patterns, contact with different animals and parasites, and access to healthcare and drugs, such as antibiotics [23,27]. The Western society microbiome presents a lower abundance overall (15–30% fewer species of microorganisms) and in specific phyla (such as *Bacteroides*—more abundant in “Western” populations, and *Firmicutes* and *Proteobacteria*—more abundant in “non-Western” populations) [18,20,21,22]. Certain species, such as bacteria from the genus Treponema, are entirely absent from the Western microbiome, most likely because of the reduced role of agriculture, gathering, and hunting [22,26].

When compared to non-human primates, such as African apes, human gut microbiome is less diverse, with reduced abundance of *Methanobrevibacter* and *Fibrobacter* and increased one of *Bacteroides* [28]. This is connected with the higher amount of meat and cooked food consumed by humans. Other factors include the size and density of human populations, the role of agriculture and the presence of N-glycolylneuraminic acid in humans, but not in apes [15,28]. The microbiomes of non-human primates are also influenced by season and habitat; however, there is a pattern called phylosymbiosis, as a result of which the composition reflects host phylogenetic relationships and is not influenced by diet. It is also observed in humans: the human gut microbiome is comparable to the one of monkeys and apes from Africa, Asia and Europe, but less similar to primates and lemurs from the Americas [29,30,31,32].

### 3.2. The Gut-Brain Axis in the 18th Century

In 1681 and 1683, Antonie van Leeuwenhoek observed bacteria—*animalcules*—first in his stool samples, and then in his saliva [33,34] (Figure 2).

However, this observation was not followed with further scientific analysis for almost 100 years, when the connection between the gastrointestinal and central nervous systems gained attention in France. Paul Joseph Barthez (1734–1806) worked there on his concept of the *principe vitale* (vital principle), published in the book named *Nouveaux élémens de la science de l’homme* (1778). The vital principle was a force responsible for the motions in essential organs. Some organs, such as muscles, were observed to lose their functions soon after denervation or cutting the blood circulation. Others, such as intestines or heart, could preserve the motion for several hours. He therefore concluded that some organs must contain a higher amount of vital principle than others [35]. A student of Barthez, Jean Charles Grimaud (1750–1789), developed this theory and divided the vital principle into the subtypes: *force mortice animale* (motor animal force) and *force mortice vitale* (vital motor force). Motor animal force was responsible for external functions, such as sensory ones. The brain was responsible for receiving external, conscious stimuli. The vital motor force was connected with nutrition, reproduction, respiration, glands functions and other activities essential for the organism to survive. It was to be regulated by *sens vital interieur* (internal vital sense), which was not controllable by the will, but rather by a center in the stomach [36]. Grimaud’s theory was an inspiration for Marie François Xavier Bichat’s vitalist theory, which indicated the crucial harmony between the gastrointestinal system and the nervous system. Vitalism differentiated living organisms and non-living objects by attributing a life-giving principle, or spirit, to the organisms. The digestion was, according to Bichat, caused by such spirit—or forces of irritability and sensibility—connected to the nervous system [8].

“The nervous sympathy” was a term coined by Robert Whytt from Scotland in 1765. When he observed a huge amount of nerve endings in the intestines, he believed that this was the system of connecting the organs together by “nervous energy” [37].

While the diet was not as widely discussed subject in the 18th century, tea played a crucial role in one of the first women’s political protest in the Colonies. In 1773, Penelope Barker and 50 other women in Edenton, North Carolina, proclaimed tea boycott in response to the passing of Tea Act, probably being inspired by the Boston Tea Party (Figure 3).

### 3.3. The Gut-Brain Axis in the 19th Century

In the 19th century, physicians adopted a more holistic approach to human health, focusing on the entire body rather than individual organs. However, they paid great attention to the gastrointestinal tract. They hypothesised that nerve endings were responsible for evoking positive emotions. On the other hand, dietary mistakes such as unhealthy food or alcohol intake could have a damaging influence on them, leading to negative emotions [1]. They believed, as Royal physician James Johnson wrote in 1827, that “*strange antipathies, disgusts, caprices of temper, and eccentricities, which are considered solely as obliquities of the intellect, have their source in corporeal disorder*” [38].

The heart was linked to positive emotions, while the stomach was associated with negative ones. Vomiting as a symptom of disgust or anxiety seemed to be a visible sign of this connection. The interpretation of symptoms was also subject to one’s gender. Women were perceived as having weaker stomachs, so they presented more often with “sinking” and fluttering heart. Men’s symptoms were often attributed to diet, and if the diet mistakes were frequent, the anxiety or irritability could be even permanent [9].

Whytt’s concept of “nervous sympathy” became popular among physicians in the 19^th^ century and the gastrointestinal tract started to be a popular literature subject. The stomach was promoted to be one of the most important organs. Terms such as “*the great abdominal brain*”, “*the sensorium of organic life*” or “*the great nervous centre*” were created and used to describe the concept of gut-brain axis in those times [39,40]. However, the colon was not the part of the scholars’ interest, as it was perceived “*as a tube that merely stored and evacuated the waste products of digestion*” [41] or “*simply a sewer canal*” [42].

John Abernethy, a London physician, published the books titled *Surgical Observations on the Constitutional Origin and Treatment of Local Diseases* (1811) and *The Abernethian Code of Health and Longevity* (1829) [43], in which he promoted the theory of “the nervous sympathy”. He disputed the connection between all somatic and mental diseases and gastric derangement, which was rooted in the neural connection between the gastrointestinal system and the mind. Digestive dysfunction was suspected as an underlying cause of the sleep and mood abnormalities, anxiety and fatigue, however, it could be prevented by avoiding highly processed food, which was typical of industrial England [44].

The epidemic of gastrointestinal disturbances that arose from urbanisation and industrialisation was claimed the most common medical problem by the Medico-chirurgical Review in 1826 [45]. In 1838, the *Dublin Journal of Medical Science* stated that stomach diseases were a national disease and “*the prime staple of the medical art*” [46]. The temperance literature of the 1840s claimed that it was a new phenomenon for the working class [47]. This observation stems from the fact that gastrointestinal abnormalities were treated among the upper class in the 18th century and were linked to overeating, overthinking, and sedentary lifestyle [48]. In the late 19th century, indigestion remained a “fashionable” disease, with an intensive advertisement for digestive drugs [49]. However, the phenomenon was observed across all social classes, which could reflect major social changes and a changing approach to health within the whole society. The physicians encouraged the English nation to eat regularly, slowly, healthy and in moderate amount, and to avoid alcohol. The most popular of those publications were *Memoirs of a Stomach* by Sydney Whyting (1853). The main character was Mr. Stomach, who was suffering from unhealthy eating habits and poor emotion management of its owner [50] (Figure 4).

A gloomy approach to this problem was published in 1861 in *Blackwood’s Edinburgh Review.* The report stated that the English were the most susceptible nation to indigestion, and that dyspeptic episodes often led to absolute despair. The statement was supported by a story of an Englishman who was physically and mentally unhealthy, and it claimed that autumn and winter would harvest multiple corpses of the suffering people [51].

The theory of “autointoxication” was connected to the dangers of constipation. It was believed that if toxins produced during digestion were not evacuated from the body early enough, they would lead to systemic poisoning, including the central nervous system. It was connected with the discovery of microorganisms in putrefy animal and vegetable samples, suspected to exist also in human colon [52]. German and French physicians, such as Charles Bouchard (1887), connected the existence of gut bacteria to autointoxication [53]. These phenomena were connected with dyspepsia and neurasthenia gastrica. The coprophagia, the act of ingesting a stool, was marked by physicians as both the symptom and the cause of mental illnesses [1,54].

The 19th century was also the time of a huge public discussion in the United Kingdom on excessive tea consumption. Though discouraged by the physicians, women of the late Victorian period often centred their diet around white bread and tea. This was at least partially connected to the necessity of leaving more nutritious food for the rest of their family [55]. The Dean of Bangor, Henry Thomas Edwards, in 1883 claimed that the tea consumption of the working class led to nervosity, hysteria and discontent, with poor control of their negative emotions. The tea was even marked as “*a dangerous, revolutionary force*”, seemed “*to be swelling into a flood of radicalism*” and lead to “*lamentable immorality*” [56]. Tea consumption was one of the popular ways to explain neuropsychiatric symptoms, such as so-called “hysteria,” in women. An example of a 32-year-old tea drinker who presented with psychoneurological symptoms leading to “hysterical fits” was published in *Freeman’s Journal* in 1872 [57]. The consumption of tea and white bread was also questioned in the context of the huge number of psychiatric asylum admissions in both Britain and Ireland in the 1890s, despite the decreasing Irish population due to emigration [58].

It is difficult to pinpoint an exact moment from which the scientists and clinicians gained interest in microbiota. However, in 1842, John Goodsir discovered so-called *Sarcina ventriculi* in a patient’s stomach, which he believed was a harmful organism and could be eradicated with an antiseptic drug [59]. Friedrich Theodor von Frerichs opposed him, believing *Sarcina ventriculi* should be classified as one of the commensal organisms found in the gastrointestinal track [60]. In the mid-1880s, the discussion about gut microbiome existed in Germany, Britain, and the United States [61,62,63]. Theodor Escherich discovered that a healthy child’s gut contains a type of bacteria also observed in the patients with diarrhoea, later named *Escherichia coli*. These discoveries led to the future discovery of other compounds of the human gut microbiome, such as *Veillonella parvula* in 1898 by Veillon and Zuber or Bifidobacterium in 1900 by Tissier [60,64].

The detailed pathological description of typical sites and types of such diseases as gastritis or ulcerations by John Abercrombie (1828) undermined the role of the term “dyspepsia” [65]. It led to isolation of one organ, such as the stomach, from the patient as a whole. The development of laboratory medicine and the discovery of molecular background of so-called “dyspepsia” isolated the stomach even more. Max Einhorn’s stomach bucket (to collect the chemicals from stomach), stomach tubes and devices using gases or liquids allowed the analysis of the gastrointestinal tract from the perspective closer to modern science [66,67]. However, both physicians and patients were sceptical and preferred “the old ways” [68].

In 1891, Nencki, Macfadyen and Ziber-Szumowa wrote a case report about a woman who had abdominal surgery due to intestine perforation and necrosis. The authors analysed the patient’s diet and stool and found that the stool contained bacteria involved in digestion [69].

### 3.4. The 20th and 21st Century

By the beginning of the 20th century, the 19th century holistic approach to gastrointestinal research was reversed, with researchers concentrating more on the stomach from an anatomical or surgical approach. Peter Down, a British psychiatrist, postulated that all patients with gastrointestinal symptoms without any attainable organic cause should be diagnosed for mental illness [70]. This would lead to the classification of all functional disturbances as being mainly caused by psychological distress. The theory of autointoxication was also discredited in the 20th century by the accessibility of anti-constipation herbal remedies and enemas, and by the popularity of such diet gurus as John Harvey Kellogg [53].

However, it does not mean that research on microbiota and the gut-brain axis froze completely throughout the 20th century. Many felt disillusioned by the limitation of the novel diagnostic and research methods. William Fenwich wrote in 1910 that it is impossible for laboratory medicine to answer the question about stomach reaction to strong emotions, “*despite the fact that experiments are supposed to have proved [old methods] to be too unscientific in origin and useless in application*” [71]. The newly emerged psychoanalysts emphasised the role of emotional state in gastrointestinal symptoms [10,11,12]. Gastrointestinal problems started to be mainly connected with stress. Gastric ulcers were thought to occur in patients with restlessness and anxiety [72]. The stress-related gastritis appeared in the background of important historical and social events, such as London Busmen’s Strike in 1937 [73]. The pattern of stress-relating ulcers was confirmed during the Second World War, when the soldiers who fought at Dunkirk and Londoners affected by air raids were observed to be prone to develop ulceration [74].

The beginning of the 20th century is connected with further advances in microbiology. In 1901, Elie Metchnikoff gave a lecture on the colonisation of the infant’s body after delivery, as it was believed that the infant was completely sterile until the moment of delivery. In 1904, he observed that certain microorganisms prevented the sour milk from rotting. He hypothesised that the same “intestinal putrefaction” in human gut, leading to ageing, could be stopped by certain species of bacteria, such as *Lactobacillus*. This led to the production of the first precursor of modern probiotics—lactobacilline pills—produced in Paris by the Le Ferment company between 1905 and 1910 [64] (Figure 5).

This time was also connected with the works of Arthur Samuel Kendall, who wrote about the connection between diet and microbiota composition. However, he also noted that it was not possible to identify all microorganisms due to limited research methods [75]. It was believed until the 1970s that the microbiota consisted of 400–500 species, but recent research estimates 1000 species using modern methods.

During World War I, Alfred Nissle observed that colonisation with certain strains of *Escherichia coli* led to resistance to dysentery among the soldiers [60].

The microscopic proof of the existence of microbiota in a rodent stomach was published in 1965 by Dubos et al. The researchers observed that while some species are found in the whole gastrointestinal tract (*Lactobacilli and Streptococci*), some are seen only in some parts (Bacteroidetes found only in the large intestine). The abundance of those three species was also relatively conservative throughout the entire life of the rats [76].

After the Second World War, the world once again turned towards reductionism. This led to frustration among many patients and physicians, who felt that while they had gained more complex diagnostic methods, they were still in the dark when it came to the nature of the gut.

In 1951, a book called *How I Cured my Duodenal Ulcer* was published by a patient named John Parr. The story follows his struggle with gastric ulcers, which were difficult to diagnose using both radiology and surgical methods and were believed to be untreatable by diet. The ulcers appeared after he served in the army during the Second World War, but no conventional or unconventional method seemed to help him. Parr eventually concluded that even the smallest amount of stress was the factor that exacerbated his illness [77].

However, the creation of an H2 receptor antagonist in the 1970s and the discovery of the molecular background of dyspepsia, as well as a connection between Helicobacter pylori and ulceration in the 1980s, reversed interest back into the direction of reductionism, but it dissuaded interest from psychological factors [78,79].

In 1992, Bocci coined the term “neglected human organ” to describe the importance and role of the gastrointestinal microbiota. He later observed that the lipopolysaccharide from Gram-negative bacteria’s cell walls could prevent the host from infectious diseases and neoplasms by immune system activation [80]. In 1998, Michael Gershon published a popular science book titled *The Second Brain*, in which the author elaborates a scientific discovery called revolutionary for its time: that the nerve cells in the gut have an independent, not brain-controlled influence on the functions of the intestine [81].

The intervention methods began to be created or rediscovered at the beginning of the 20th century. In 1910, I.O. Wilson launched rectal delivery of faecal microbiota transplantation [82]. The human stool was effectively used as a treatment for Clostridium difficile infection in 1958 and 1981 [83,84]. The topic of microbiota transplantation involving the vaginal microbiota from mother to infant born via caesarean section is an emerging area of research [85].

The 20th century is associated with another new term, “probiotics”, coined by Ferdinand Vergin in 1954 and defined by Daniel Lilly and Rosalie Stillwell in 1965 [86]. The final form of the definition was approved by the World Health Organisation in 2001.

At the end of the 20th century, Joshua Ledenberg popularised the term “microbiota”, which helped with justifying further research within this field. However, it was in the 21st century that the term “human microbiome” was truly conceptualised. In 2007, the HMP program began. The main goal of the HMP was to identify the composition of the human microbiome and the role of each type of microorganism. Whole genome sequencing techniques enabled its development. The project was a success: over 200 bacterial species were identified and linked to their function in the human organism, some of which were unknown before the project [87]. In 2008, the European Commission launched a new international consortium, known as MetaHIT. The goal of this project is to identify the genomes of gut microbiota and to create methods of analysing the microbiota of an individuals’ gut; both of the tools determine the species or types of microorganisms and bioinformatic tools for data analysis and storing. MetaHIT focuses on the role of the microbiota in inflammatory bowel disease and obesity [13].

The microbiome appears to be a promising new therapeutic target, especially in the context of the discovery of probiotic bacteria serving as a potential treatment method for neuropsychiatric symptoms [88]. Those types of bacteria are called “psychobiotics” and include many species [89]. These microorganisms provide the patient with neurotransmitters and other substances with an influence on the central nervous system. Previous research has demonstrated its positive impact on patients with depression, anxiety, stress, sleep disturbances, autism, Parkinson’s disease, and Alzheimer’s disease [90]. Anderson, Cryan, and Dinan suggest that psychobiotics could be a solution to modern Western societies epidemics of depression [52].

## 4. Conclusions

The history of microbiota and the gut-brain axis is complicated and includes many twists and turns. The concept of microbiome seems very complex and equivocal. The changes in approaches and fashions in the clinical and scientific approach to the gut-brain axis have led to many unexpected discoveries on the one hand, and sometimes to running in circles on the other hand. The history of microbiota research is still ongoing and this is a very fast developing field of science, which could be both optimistic and interesting. Taking into consideration both somatic and laboratory results, as well as the psychological sphere of the patient, the current holistic approach seems to be promising for future research projects and gives hope for a better understanding the influence of gut microbiota on the central nervous system.

## Figures and Tables

**Figure 1 pathogens-11-01540-f001:**
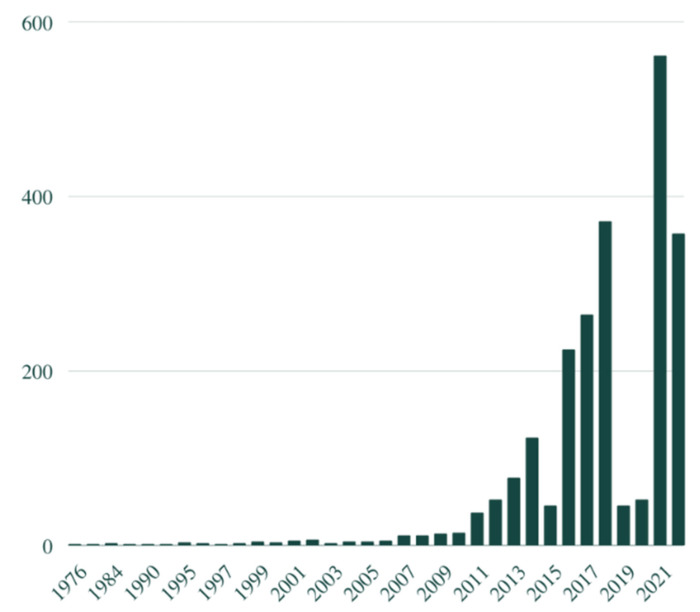
The number of articles published on microbiota history or microbiome history or gut-brain axis history. Graph created with Canva tool.

**Figure 2 pathogens-11-01540-f002:**
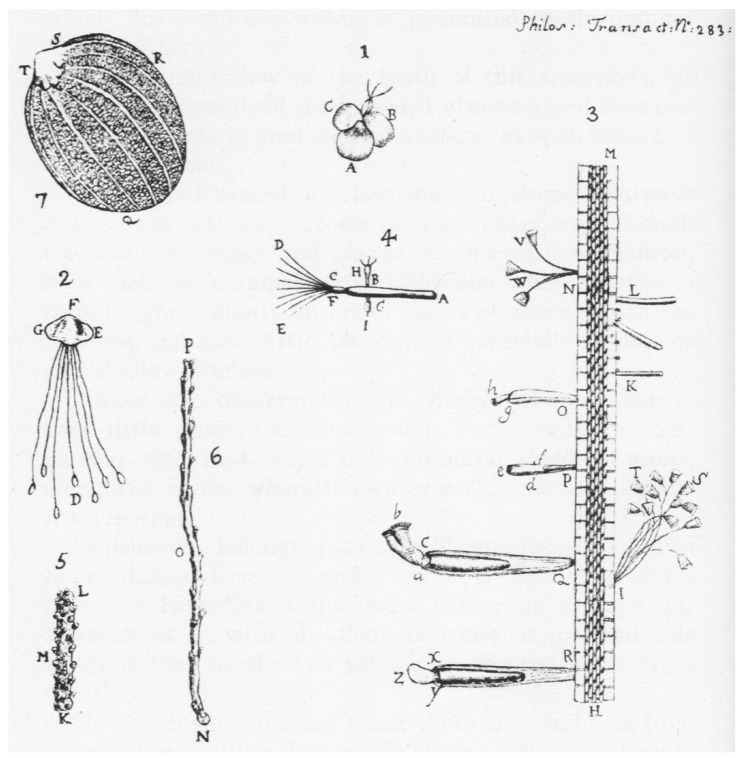
Illustrations to Leeuwenhoek’s Letter on the animalcules. Dobell C (1960) Antony van Leeuwenhoek and his “little animals”. Dover Publications, New York (Public Domain).

**Figure 3 pathogens-11-01540-f003:**
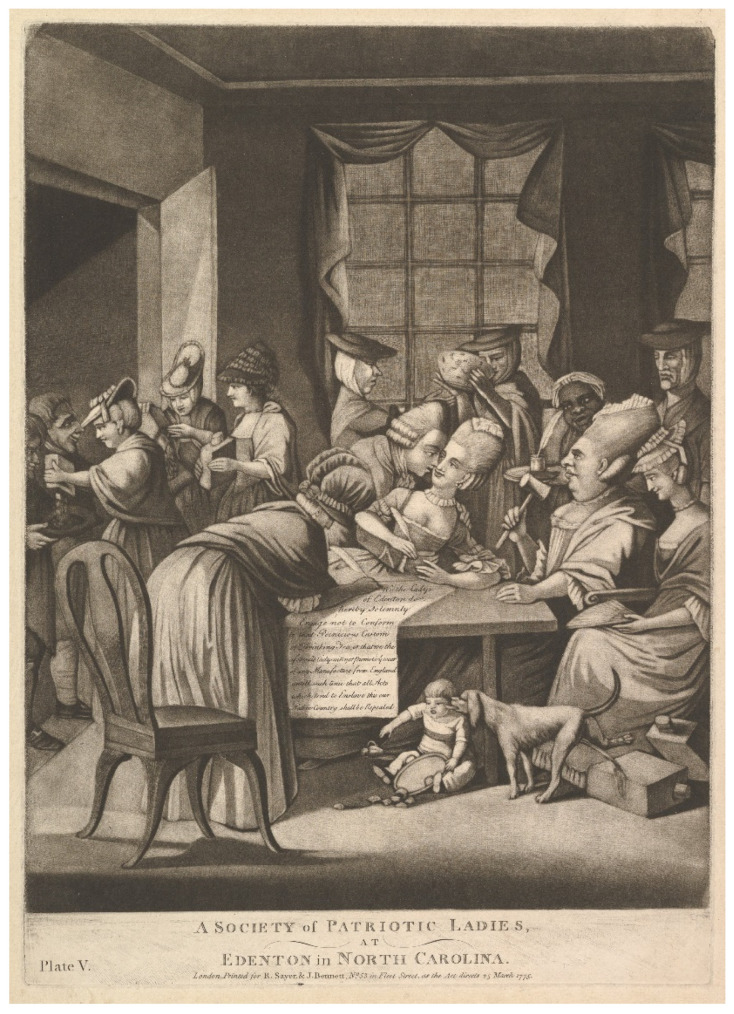
A British satirical drawing, published in response to the Edenton female tea boycott. Dawe P (1774). A society of patriotic ladies, at Edenton in North Carolina. R. Sayer & J. Bennett, London (Public Domain).

**Figure 4 pathogens-11-01540-f004:**
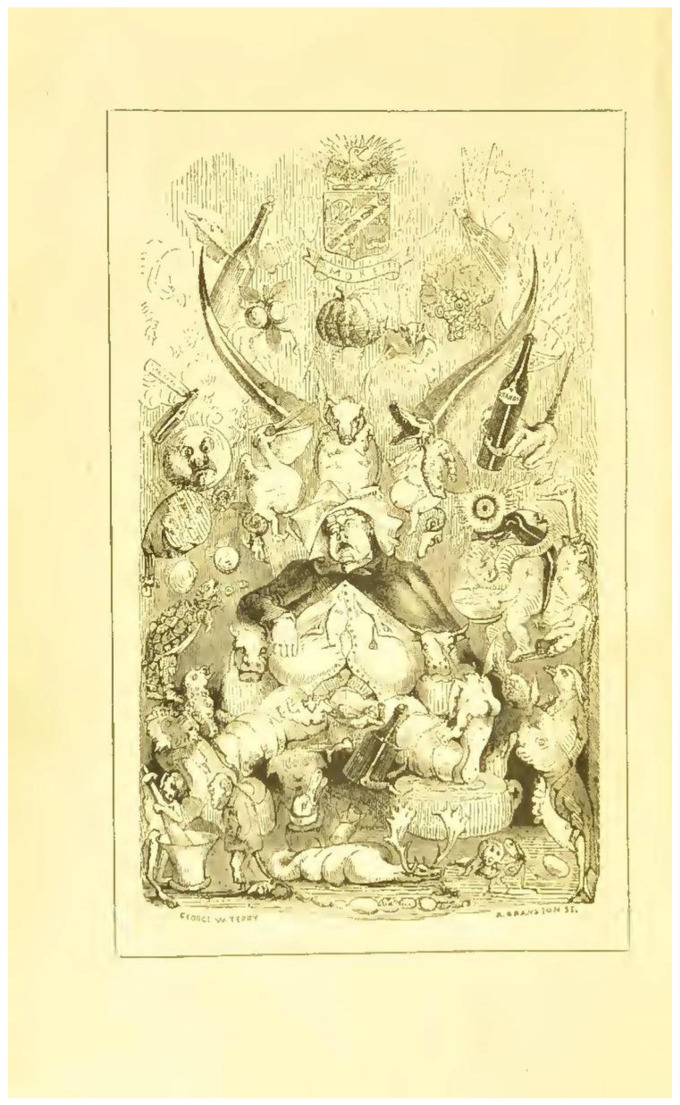
Mr. Stomach, the main character from “Memoirs of a Stomach” (Public Domain).

**Figure 5 pathogens-11-01540-f005:**
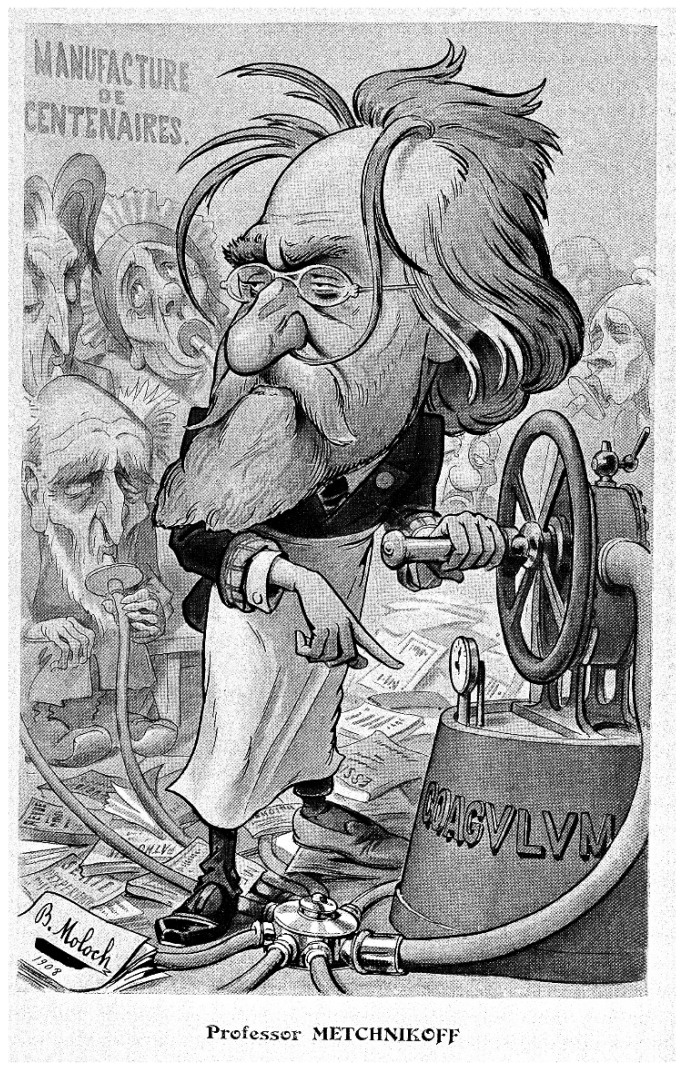
The satirical drawing of Elie Metchnikoff, depicting his attitude to probiotics as panacea. Moloch H (1910). Le Professeur Metchnikoff. Revue Chanteclair, Paris (Public Domain).

## Data Availability

Not applicable.

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
