# Peer review of "The History of the Intestinal Microbiota and the Gut-Brain Axis"

_pathogens, 2022, doi:10.3390/pathogens11121540_

Round 1

Reviewer 1 Report

Lewandowska-Pietruszka et al., did an interesting review of the history of gut-brain axis. Please see my minor comments below:  

1.      Line 25, …”in both scientific and popular science literature” is not needed.

2.      Line 31-32, citation

3.      Line 37-42, please add citations at the end of each sentence because they are claims.

4.      Figure 1, could the authors make this figure black-white? There is no need to use light blue as it is a simple barplot. To MDPI editorial office, this is too large of a figure for its content in proof.

5.      I suggest removing the results section and combining it into the discussion as Results and Discussion. The 4 figures in the results are abrupt without any explanations until discussion. It’s also hard for readers to understand with the historical figures that come before the discussion part.

6.      I recommend removing “short” from the title. History is not short…

Author Response

Dear Reviewer. Thank you very much for your kindness and insightful review.

I'll start from the end. The story is not short, I agree. The word "short" has been removed from the title. Thank you for suggestion. 

And the rest: 

1.   Line 25, …”in both scientific and popular science literature” is not needed. - It was removed

2.      Line 31-32, citation - added

3.      Line 37-42, please add citations at the end of each sentence because they are claims - added

4.      Figure 1, could the authors make this figure black-white? There is no need to use light blue as it is a simple barplot. To MDPI editorial office, this is too large of a figure for its content in proof. - The figure has been scaled down and changed to a black and white variant.

5.      I suggest removing the results section and combining it into the discussion as Results and Discussion. The 4 figures in the results are abrupt without any explanations until discussion. It’s also hard for readers to understand with the historical figures that come before the discussion part. - We combined "results and discussion". I hope this makes it easier to read the article.

Thanks again for the quick review of the article.

On behalf of the authors: Katarzyna Mazur-Melewska

Reviewer 2 Report

Dear Authors,

This is a very interesting review as it analyzes both scientifically and historically many aspects of the relationship between the intestinal microbiota and the gut-brain axis.

I have some minor comments:

Introduction:

Line 26: missing “the” before discovery

Lines 37-42: perhaps there are references missing for supporting the findings of 19th, 20th and 21st centuries. There is one reference (5) in this paragraph but it is in Polish and since, I don’t know Polish, I’m not sure if this reference supports the aforementioned findings.

Each Figure should be placed after its first mention in the text, according to the journal’s instructions.

In the paragraph “The gut-brain axis in the 18th century”, I believe that the authors should further expand and analyze the Marie François Xavier Bichat’s vitalist theory, as little to none is reported on this.

In the paragraph “The gut-brain axis in the 19th century”, there are some mentions of 18th century that according to my opinion, should be moved to the previous paragraph that analyzes the findings of the 18th century: Lines 181-190 and 246-248.

There is a missing “.” in line 255.

In the paragraph “The 20th and 21st century”, there is a reference missing: Line 282.

Lines 340-344, there are missing references.

Author Response

Dear Reviewer,

Thank you very much for the quick but very thorough evaluation of the article.Let me answer in turn.

- Line 26: missing “the” before discovery - added

- Lines 37-42: perhaps there are references missing for supporting the findings of 19th, 20th and 21st centuries. There is one reference (5) in this paragraph but it is in Polish and since, I don’t know Polish, I’m not sure if this reference supports the aforementioned findings. - The references were added.  The order of references has been changed

- Each Figure should be placed after its first mention in the text, according to the journal’s instructions. - The location of figures has been improved

- In the paragraph “The gut-brain axis in the 18th century”, I believe that the authors should further expand and analyze the Marie François Xavier Bichat’s vitalist theory, as little to none is reported on this. - dopisane

- In the paragraph “The gut-brain axis in the 19th century”, there are some mentions of 18th century that according to my opinion, should be moved to the previous paragraph that analyzes the findings of the 18th century: Lines 181-190 and 246-248. - Opinions for the 18th century have been shifted as suggested.

- There is a missing “.” in line 255. - added

- In the paragraph “The 20th and 21st century”, there is a reference missing: Line 282. - The reference is added

- Lines 340-344, there are missing references. - dodane + usunęłam fragment o stolcu wielbÅ‚Ä…dów - The references were added. The sentence: "During the Second World War, the camel stool was used among soldiers in Africa to treat dysentery." The authors found new data that are inconsistent with earlier ones. The topic needs updating.

I hope we were able to improve the article.

In behalf of authors:

Katarzyna Mazur-Melewska
